# Ultrasound-Guided Compression Method Effectively Counteracts Russell’s Viper Bite-Induced Pseudoaneurysm

**DOI:** 10.3390/toxins14040260

**Published:** 2022-04-06

**Authors:** Subramanian Senthilkumaran, Stephen W. Miller, Harry F. Williams, Rajendran Vaiyapuri, Ravi Savania, Namasivayam Elangovan, Ponniah Thirumalaikolundusubramanian, Ketan Patel, Sakthivel Vaiyapuri

**Affiliations:** 1Emergency Department, Manian Medical Centre, Erode 638001, India; maniansenthil76@gmail.com; 2The Poison Control Center, Children’s Hospital of Philadelphia, Philadelphia, PA 19104, USA; millste4@isu.edu; 3Research and Development Department, Toxiven Biotech Private Limited, Coimbatore 641042, India; harry@toxiven.com (H.F.W.); raj@toxiven.com (R.V.); 4School of Pharmacy, University of Reading, Reading RG6 6UB, UK; r.savania@reading.ac.uk; 5Department of Biotechnology, School of Biosciences, Periyar University, Salem 636011, India; elangovannn@gmail.com; 6Research Department, Trichy SRM Medical College Hospital & Research Centre, Trichy 621105, India; ponnian.tks@gmail.com; 7School of Biological Sciences, University of Reading, Reading RG6 6UB, UK; ketan.patel@reading.ac.uk

**Keywords:** pseudoaneurysm, snakebite envenomation, Russell’s viper, ultrasound-guided compression, ulnar artery, colour Doppler imaging

## Abstract

Russell’s viper (*Daboia russelii*), one of the ‘Big Four’ venomous snakes in India, is responsible for the majority of snakebite-induced deaths and permanent disabilities. Russell’s viper bites are known to induce bleeding/clotting abnormalities, as well as myotoxic, nephrotoxic, cytotoxic and neurotoxic envenomation effects. In addition, they have been reported to induce rare envenomation effects such as priapism, sialolithiasis and splenic rupture. However, Russell’s viper bite-induced pseudoaneurysm (PA) has not been previously reported. PA or false aneurysm is a rare phenomenon that occurs in arteries following traumatic injuries including some animal bites, and it can become a life-threatening condition if not treated promptly. Here, we document two clinical cases of Russell’s viper bites where PA has developed, despite antivenom treatment. Notably, a non-surgical procedure, ultrasound-guided compression (USGC), either alone, or in combination with thrombin was effectively used in both the cases to treat the PA. Following this procedure and additional measures, the patients made complete recoveries without the recurrence of PA which were confirmed by subsequent examination and ultrasound scans. These data demonstrate the development of PA as a rare complication following Russell’s viper bites and the effective use of a simple, non-surgical procedure, USGC for the successful treatment of PA. These results will create awareness among healthcare professionals on the development of PA and the use of USGC in snakebite victims following bites from Russell’s vipers, as well as other viper bites.

## 1. Introduction

Snakebite envenomation (SBE) can produce a myriad of clinical effects ranging from local symptoms such as necrosis, swelling and muscle damage at the bite location to serious systemic effects including paralysis with respiratory failure, cytotoxicity, kidney failure, and coagulation abnormalities (e.g., haemorrhage and consumption coagulopathy) [1,2]. SBE is responsible for a large number of deaths and permanent disabilities disproportionately in rural and impoverished areas of South and Southeast Asia, Africa and South America [3,4]. Due to the lack of reliable statistics, the estimates of SBE-induced deaths and disabilities vary widely. However, SBE is estimated to cause as many as 125,000 deaths and around 500,000 disabilities annually worldwide [5,6,7]. While many of the clinical aspects of SBE are well known, manifestations such as pseudoaneurysm (PA) following SBE are rare. These rare complications are likely to prolong hospital stay and increase the treatment costs for the victims. Therefore, it is important to thoroughly investigate SBE-induced rare complications and develop appropriate management strategies to tackle these issues.

PA is different from true aneurysm, which develops due to weak arterial wall and subsequent elongation of a balloon-like structure in specific regions [8]. PA develops following a disruption in the arterial wall due to trauma or other similar conditions. Due to the arterial pressure, blood within the vessel leaks into the tissues surrounding the damaged artery. Subsequently, a perfused sac is formed, and it communicates with the arterial lumen. Moreover, this perfused sac is contained by the media or adventitia, or soft-tissue structures surrounding the injured blood vessel [9]. The most common aetiology of PA is iatrogenic arterial puncture often following a catheterisation for diagnostic or therapeutic procedures and surgical procedures such as a percutaneous biopsy [10]. Additional causes may include infection, inflammation, vasculitis, or drainage. The development of PA after injuries sustained from animal bites (cats and dogs) as well as a stingray puncture wound have also been reported [11,12]. PA can lead to more serious complications that require urgent non-invasive interventions and/or surgeries, including amputation [13,14]. Here, we demonstrate the development of PA in two Russell’s viper bite victims, despite antivenom treatment and the successful use of a non-invasive, ultrasound-guided compression (USGC) method to treat these complications. These data demonstrate the impact of Russell’s viper bites in inducing such rare complications, and the robust use of a non-surgical procedure to tackle this issue. Notably, this article creates more awareness about Russell’s viper bites and their complications in SBE victims.

## 2. Results

### 2.1. Clinical Presentation of Victims Who Developed PA

The first patient, a 50-year-old female, was bitten by a Russell’s viper in her left cubital fossa while doing agricultural work on a farm. The offending snake was identified as a Russell’s viper by the patient and their relatives, and they also displayed a photo of the offending snake to the clinicians and a herpetologist who verified it to be a Russell’s viper (Figure 1A). The patient was first taken to a local primary hospital within one hour of the bite and she received 200 mL of polyvalent antivenom produced against ‘Big Four’ snakes (Russell’s viper, cobra, krait and saw-scaled viper) of India to manage the abnormalities in coagulation profile. The systemic envenomation effects were improved following the antivenom administration. There were no further signs of envenomation on day 2 and 3. However, on day 4, she was referred to the emergency department and upon location examination, she found to have a large pulsatile mass of approximately 4 cm × 3 cm in her left cubital fossa (Figure 1B). The mass was rubbery, firm to fluctuant, tender and had a palpable thrill with an audible bruit. A mild drainage was also present but there was no evidence of distal blood flow compromise. No other signs of envenomation were evident including skin ulceration or necrosis, regional lymphadenitis or systemic envenomation effects indicating the effectiveness of timely antivenom treatment. She was conscious, well oriented, afebrile, haemodynamically stable and maintaining adequate saturation in room air. The rest of the physical examination was unremarkable and her haematologic, metabolic, biochemical and coagulation profiles were within the normal limits.

The second patient was a 45-year-old female, who was bitten by a Russell’s viper in her right cubital fossa when cutting grass in her farmland to feed their cattle. The envenomation effects were conservatively managed initially with a local compression. She subsequently developed coagulopathy within two hours following the bite and therefore, was administered 250 mL of polyvalent antivenom at a local primary hospital. The patient was discharged on day 5 as she did not have any further signs of envenomation, and all her haematologic and metabolic parameters were normal. However, she developed progressive swelling around the bite site in right cubital fossa from day 5. The hospital clinician in the local hospital advised that it would disappear slowly following discharge from the hospital. On day 6, she reported developing numbness in her hand and tingling sensation in her fingers which was especially pronounced in thumb and adjacent fingers. Additionally, she described constant aching pain which was aggravated by movement of the elbow joint. As there was no improvement, two weeks following the bite she was brought to the emergency department. Upon presentation, she was conscious, well oriented, afebrile, haemodynamically stable and maintaining adequate saturation in room air, and no other systemic envenomation effects observed. However, physical examination revealed a 2.4 cm × 3 cm pulsatile mass in her right antecubital fossa that was tense, rubbery, firm to fluctuant and ecchymotic (Figure 2A). There was no evidence of compromised distal blood flow or local signs of envenomation such as necrosis, tissue damage, or regional lymphadenitis. All her haematological and other vital parameters were within the normal limits.

### 2.2. Ultrasound Diagnosis Confirms the PA in Ulnar Artery of Victims

In both cases, diagnosis was made using multiple ultrasound techniques. Initially, the first patient was assessed using a grey-scale ultrasound scan (Philips, India) which showed a pulsatile cystic structure in proximity to the respective ulnar artery in the left upper extremity (Figure 1C). A pulsed Doppler spectral analysis revealed a “to and fro” pattern in the aneurysmal neck (Figure 1D). Further analysis using colour Doppler imaging showed a swirling blood flow indicating the “yin-yang” sign (Figure 1E). In the second patient, the colour Doppler imaging was used, and this confirmed the presence of the “yin-yang” sign, demonstrating the turbulent blood flow (Figure 2B). Based on these findings, both patients were diagnosed with PA on the ulnar artery. Both patients confirmed that they have no other known pre-existing health conditions.

### 2.3. USGC Method Effectively Treats PA

After discussing the risks and benefits of treatment with the first patient and her family members, the decision was made to perform a non-invasive/non-surgical USGC method to treat the PA. The patient received three cycles of USGC, each for 10 min in duration on day 4, 5 and 6 following the bite. Both clinical observations and Doppler ultrasound reassessment two weeks later demonstrated a reduction in size of the PA without any evidence of compromised parent artery patency. For the second patient, various treatment options were discussed, and surgical options were dismissed by the patient and their family members. Hence, the USGC method was initiated, however, the patient was unable to tolerate the pain. Therefore, thrombin (three injections with a total of 10,000 IU in saline) was administered in the PA prior to using USGC for three times, each 10 min in duration on the same day. There was no evidence of bleeding after her treatment, and she completely recovered. An ultrasound scan confirmed complete thrombosis of the PA with a normal parent artery and blood flow. For both patients, no other special treatments were provided except standard analgesics to manage their pain. These data demonstrate the successful application of USGC in the clinical management of Russell’s-viper-bite-induced PA with/without the infusion of thrombin.

## 3. Discussion

In addition to systemic envenomation effects such as coagulation abnormalities, kidney damage and neurological complications, Russell’s viper bites result in severe local envenomation effects including pain, ecchymosis, swelling and bleeding at the bite site as well as lymphadenopathy and muscle damage [15,16]. Notably, various rare complications such as priapism [17] and splenic rupture [18] have been observed following Russell’s viper bites in victims. Although aneurysm or PA has not been previously reported in Russell’s viper victims, a few reports of these complications are available for other snakebites. A 65-year-old male was reported to develop aneurysm of the radial artery after envenomation at their wrist by *Deinagkistrodon acutus* [19]. Despite suitable antivenom treatment, the patient developed a pulsatile mass near their radial artery that ruptured and, therefore, was surgically removed. In addition, multiple debridement as well as a skin graft were performed in this patient without any success. Finally, the arm was amputated 15 days following the bite as there were no other options available. Another case of PA was reported in a 61-year-old male patient who was also bitten by a *Deinagkistrodon acutus* on the forearm [20]. The patient received a fasciotomy to manage the compartment syndrome. Upon debridement 10 days later, two PAs were discovered on the ulnar artery. These were surgically removed and then the patient’s condition improved. The patient was discharged after more than a month of hospitalisation. An additional case of an aneurysm on the posterior tibial artery was reported in 1982 following a snakebite although we could not obtain additional details as the article was in German and not freely available [21].

Viper venoms are known to contain proteolytic enzymes such as metalloproteases and serine proteases as well as phospholipase A_2_ [1,2]. Some of these enzymes have pro- or anti-coagulant activities and others specifically, metalloproteases cause destruction of multiple basement membrane components (e.g., collagen) in the vasculature. Some metalloproteases are known to cause extensive haemorrhage by damaging extracellular matrix in blood vessels [1,22]. Metalloproteases and serine proteases are present in the venom of Russell’s vipers from different locations [23,24]. Venom-derived vascular apoptosis-inducing proteins, as well as certain L-amino acid oxidases may induce apoptosis in the endothelial cell lining of the blood vessels which can also contribute to further damage [25,26]. Phospholipase A_2_s present in the venom of the Russell’s viper are known to induce inflammatory responses producing reactive oxygen species and synthesis of various cytokines [24]. It is important to note that certain viper venom components such as hyaluronidases aid the spread of other venom components which may amplify their deleterious effects [27,28]. Several non-enzymatic proteins such as snaclecs can affect platelet function, and thus leading to bleeding. While the deleterious effects of venoms, specifically viper venoms including Russell’s viper, on blood vessels are not surprising, the development of aneurysm or PA warrants additional investigation. We cannot rule out the possibility of other underlying health conditions in the development of PA in these two patients.

Neither of the patients displayed local effects at the time they were presented to the emergency departments aside from the PA. This suggests that their previous antivenom treatment was indeed effective in managing their envenomation effects. They presented with PA 3 and 14 days, respectively, after being bitten by Russell’s vipers. Cubital fossa is a small triangular region in the anterior surface of the elbow, and is where the brachial artery branches into ulnar and radial arteries [29]. It is possible that the penetration of the fangs into the local tissues surrounding ulnar arteries induced localised physical trauma which resulted in the development of the PA despite antivenom treatment. The penetration of fang(s) directly into the ulnar artery may have damaged the vessel and served as a mechanism for direct intra-arterial injection of the venom. This might explain the development of systemic coagulopathy in the absence of local effects or lymphadenitis in these patients. The direct intra-arterial injection of snake venoms has not been previously reported to the best of our knowledge. Alternatively, the PA might have developed due to the deposition and local actions of venoms in proximity to the involved artery.

Our study shows that the diagnosis of PA can be made based on clinical evaluation of the local affected region and ultrasound scans which are both affordable and readily available in most clinical settings. Triplex ultrasound combining B mode, colour Doppler, and spectral analysis may also be used to ascertain the nature of PA and blood flow in the affected region. The three characteristics of PA that were clearly apparent on ultrasound scans were an expansile pulsatile mass, ‘yin-yang’ sign (turbulent flow) and haematoma with variable echogenicity. The pathognomonic sign of PA is demonstration of a “to and fro” spectral waveform in the neck of the PA [30]. These findings suggest a communication, or blood flow between the artery itself and PA sac via the neck or stalk [31]. However, angiographic study can also be used when necessary to exclude thromboembolic diseases and to understand the anatomical status of vessels including aneurysm or PA [32].

USGC is a common method used to treat aneurysm and PA in other clinical settings [33,34] although, this has not been used in SBE victims previously. In the first patient, three sessions (each 10 min in duration) of non-invasive USGC were successful in treating the PA. However, the second patient was unable to tolerate the USGC and therefore prior thrombin injection was necessary to clot the blood and then apply USGC. Although thrombin administration helped in this patient, it may cause adverse effects such as distal embolism, allergic reaction to thrombin, and neuralgic symptoms from nerve compression [35]. When considering less-invasive treatments, some patients will recover with simple application of compression bandage over the lesion site. Surgery is often reserved for cases of PA with evidence of distal ischemia, failure of USGC, infection or necrosis of the PA, or threat of rupture [36,37].

The clinical presentation of these patients was similar in terms of age, coagulopathy, use of antivenom and location of bite on the body (one of right and another on left). Similarly, the treatment and diagnostic methods used were the same. The time of onset of the PA differed in each case ranging from three days to two weeks. The patient who developed the PA after three days was asymptomatic while the patient who arrived 14 days post- envenomation complained of numbness, pain and a tingling sensation in the hand distal to the bite. Although the precise mechanisms behind Russell’s-viper-bite-induced PAs are unclear at this stage, this study demonstrates the possibility of developing a PA following SBE and the potential use of a simple, non-invasive USGC approach to tackle this issue. Further studies are required to unravel the mechanisms through which Russell’s viper venom components induce PA in humans.

## 4. Conclusions

Russell’s-viper-bite-induced PA has not been previously documented although the bites from this snake are known to induce several rare complications. Here, both the patients received prompt antivenom treatment which helped to manage their envenomation effects. However, later they both developed PA at different time points following the bites. Health care professionals who treat SBE victims should be aware that a pulsatile mass that develops days, weeks, or months near the bite site may be related to the destructive effects of various venom components or mechanical injury due to fangs. Patients and caregivers should receive instructions upon discharge to follow up with their primary care providers and promptly report the development of any lesions or abnormalities after an injury. Initial evaluation of the lesion may include ultrasound scans of the area to determine if turbulent “to and fro” blood flow into and out of the lesion is present. This positive “yin-yang” sign may indicate that the lesion is a PA and can be treated accordingly with consideration of the interventions discussed in this article.

## 5. Methods

The data (including photographs and ultrasound images) were collected following written informed consent from the patients in both cases. They also provided permission to publish these data in scientific journals. The patients were treated using standard protocols, and all the clinical diagnosis was performed using standard equipment available within the treating hospital. The standard grey-scale ultrasound scan was used to analyse the pulsatile cystic structure of PA. The Doppler spectral analysis was used to confirm the “to and fro” pattern in the aneurysmal neck. The colour Doppler imaging ascertained the presence of swirling blood flow, indicating the “yin-yang” sign.

## Figures and Tables

**Figure 1 toxins-14-00260-f001:**
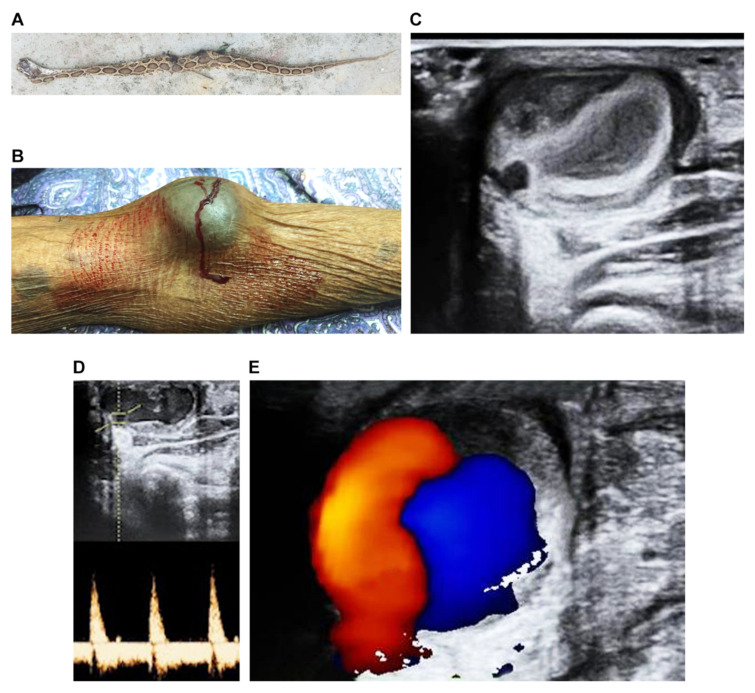
**The diagnosis of pseudoaneurysm in first patient following a Russell’s viper bite.** (**A**) The offending snake, which was identified as a Russell’s viper. (**B**) A large swelling and pulsatile mass on the left cubital fossa of the patient. (**C**) Grey scale ultrasound image confirms the cystic mass in cubital fossa. (**D**) Pulsatile mass near ulnar artery (top) and ‘to and fro’ pattern (bottom) revealed by pulsed Doppler spectral analysis. (**E**) Colour Doppler imaging reveals the ‘yin-yang’ sign of turbulent blood flow.

**Figure 2 toxins-14-00260-f002:**
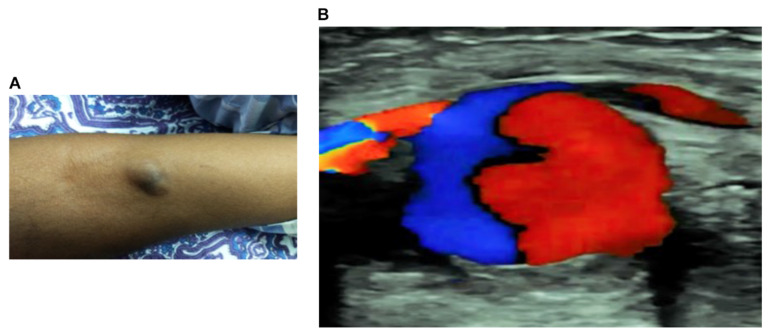
**Pseudoaneurysm in the second patient following a Russell’s viper bite.** (**A**) swelling and pulsatile mass on the right cubital fossa of the patient. (**B**) A colour Doppler imaging reveals the presence of ‘yin-yang’ sign that represents turbulent blood flow.

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
