# Peer review of "Ultrasound-Guided Compression Method Effectively Counteracts Russell’s Viper Bite-Induced Pseudoaneurysm"

_toxins, 2022, doi:10.3390/toxins14040260_

Round 1

Reviewer 1 Report

This is a case report espousing the value of an ultrasound guided compression against Russells viper bite-associated pseudoaneurysm. The authors have used two case examples to highlight the potential value of this procedure for treatment of this rare antivenom-insensitive disorder consequent to snakebite envenomation. While it is difficult to be certain that the treatment was effective or that the patients recovered independent of the treatment, the clear evidence for pseudoaneurysm and the relatively short time-frame for recovery after the procedure is encouraging. This is especially so, when compared to worse outcomes for two reports of patients with snakebite associated  pseudoaneurysms that were surgically treated. In one case this led to amputation and in the other a greater than 1 month stay in hospital.

Page 2 Line 59 Change “Notaly” to “Notably”

Figures 1 & 2: Include scale bars where possible

Author Response

We sincerely thank the reviewer for their excellent positive feedback for our manuscript. 

Page 2 Line 59 Change “Notaly” to “Notably”

We have corrected this. 

Figures 1 & 2: Include scale bars where possible

We thank the reviewer for this important suggestion. However, the equipment that we used in this study do not provide the scale bars, so we were unable to include them in the images. 

Reviewer 2 Report

The article is well written and describes a rare consequence of Russel’s bite, the pseudoaneurysm. It advises healthcare professionals to be aware of this possible complication and suggests its non-invasive treatment with ultrasound-guided compression. I believe the data shown is of interest to the readers of Toxins journal and I recommend the article to be accepted after the following minor issues are addressed:

In Methods, please include how PA was diagnosed and treated.

I found a few English mistakes and typos that should be corrected.

In abstract, line 4, apposition should be separated by commas. Therefore a comma is missing between “India” and “is”.

Two full stops in line 87.

In lines 202&204, use the abbreviation of pseudoaneurysm.

Few sentences contain repetition of the same word.

Author Response

We sincerely thank the reviewer for their constructive and positive feedback for our manuscript. We have corrected the manuscript in line with the reviewer's comments and shown them in highlighted texts in the revised manuscript. 

In Methods, please include how PA was diagnosed and treated.

We thank the reviewer for this important suggestion, and have included this in the methods section of the revised manuscript. 

I found a few English mistakes and typos that should be corrected.

In abstract, line 4, apposition should be separated by commas. Therefore a comma is missing between “India” and “is”.

We have corrected this. 

Two full stops in line 87.

We have corrected this. 

In lines 202&204, use the abbreviation of pseudoaneurysm.

We have corrected this. 

Few sentences contain repetition of the same word.

We have corrected this where possible by proof-reading the manuscript.